# Connectivity-based Token Condensation for Efficient Vision Transformer

## Abstract

The high computational cost of vision transformers blocks their implementation on resource-limited devices such as mobile phones. Reducing the number of tokens can significantly accelerate the inference process and save computational resources. Most of the existing token pruning methods focus on evaluating token's importance and discard the unimportant tokens directly, which incur significant information loss. A few methods suggest ways focusing on merging while directly partition tokens into two parts by random or odd/even partition, which do not consider carefully how to select tokens. In this paper, we propose a new token condensation method based on the connectivity between tokens. Different from the previous methods, we gradually condense the large number of tokens by selection and fusion. The most representative tokens are selected and the others are separately fused into them. Extensive experiments are conducted on benchmark datasets. Compared with the existing methods, our method achieves higher accuracy with lower computational cost. For example, our method can reduce 50% FLOPs of DeiT-S without accuracy degradation on ImageNet dataset.

## 1 Introduction

Recently, vision transformers have been widely used in various tasks, including object recognition (Touvron et al., 2020), image processing (Chen et al., 2021), and video analysis (Kim et al., 2018). Compared with convolutional neural network (CNN), the vision transformer introduces less inductive bias, which has greater potential to extract information from more training data and generalize well to more diverse vision tasks (Dosovitskiy et al., 2021; Han et al., 2022; Tang et al., 2021; Touvron et al., 2020). However, these transformer-based models have hundreds of millions of parameters and expensive computational cost, which are hard to be implemented on resource-limited devices like mobile phones. Many real-time application scenarios also require efficient models with low inference latency. Thus, model compression technologies have important application value for transformer-based models. The mainstream techniques contain pruning (Tang et al., 2022), weight decomposition (Jaderberg et al., 2014) and quantization (Liu et al., 2021b), *etc*.

Token pruning is one of those effective compression methods. Generally, token pruning methods retain important and informative tokens while discard redundant ones by comparing their importance (Kong et al., 2022; Yin et al., 2022; Wang et al., 2022; Rao et al., 2021; Meng et al., 2022; Tang et al., 2022; Fayyaz et al., 2022). However, only discarding tokens will incur performance loss as even those less important tokens may also contain certainly helpful information of the given task. A few methods (Bolya et al., 2023; Heo et al., 2023) consider to reduce the number of tokens by merging similar tokens. However, these methods only focus on merging process with simple direct relationships between image tokens. They just directly partition tokens into two parts by random or odd/even partition, which would result in the two tokens that originally need to be retained separately having to be merged together. They also overlook the potential influence of other tokens on the relationships between two tokens when analyzing their interactions.

In this paper, we propose a new token condensation method based on the connectivity between tokens. Instead of directly discarding tokens, we gradually condense the large number of tokens by selection and fusion. The most representative tokens are selected and the others separately are fused into them. To convey the relationship between different tokens, we construct a graph whose nodes are tokens. Based on the graph, we present a metric *Connectivity* to reflect both the direct and indirect

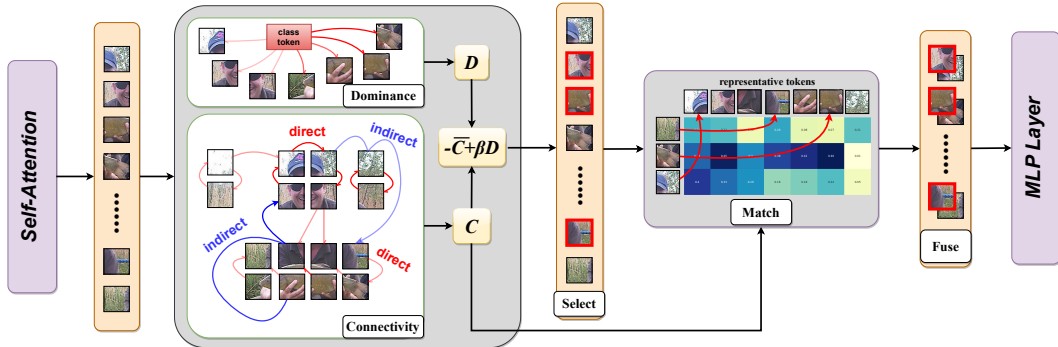

Figure 1: Overview. Firstly, we calculate the connectivity $\mathcal{C}$ and dominance $\mathcal{D}$ in order to obtain the representative tokens. Then, we match the unrepresentative tokens to representative ones. Finally, we fuse the unrepresentative tokens into representative ones.

relationships between tokens. Especially the indirect relationship (dubbed as high-order connectivity) can capture the global information to help measure the relationships among tokens, which contributes a lot both in selection phase and fusion phase. Besides connectivity, a variable, $Dominance$, is also defined to help reflect how a token is relevant to the given task. With the two variables, the representative tokens can be accurately selected and other tokens are fused into them. The pipeline is shown in Figure 1.

## 2 RELATED WORK

**Vision Transformer**    (Dosovitskiy et al., 2021) firstly introduces a pure transformer-based method to solve vision tasks. Since then, more and more transformer-based methods have emerged (Yuan et al., 2021; Han et al., 2021; Bao et al., 2021; Touvron et al., 2020; Yu et al., 2022). Following them, more and more efficient transformer methods have been proposed. Some try to prune features (Meng et al., 2022; Voita et al., 2019), and some try to infuse domain-specific modules (Mehta & Rastegari, 2022; Graham et al., 2021; Liu et al., 2021a; 2022). Besides, some works focus on token pruning (Rao et al., 2021; Meng et al., 2022; Kong et al., 2022; Xu et al., 2022; Wang et al., 2022).

**Token Pruning**    The general token pruning methods in vision transformer tasks are to prune by selecting useful tokens and discarding useless ones (Rao et al., 2021; Yin et al., 2022; Meng et al., 2022; Tang et al., 2022; Fayyaz et al., 2022; Kong et al., 2022; Wang et al., 2022; Liang et al., 2022; Tang et al., 2022). For instance, (Rao et al., 2021) inserts several lightweight modules for token selection, and combines Gumbel-Softmax to achieve end-to-end training. (Tang et al., 2022) proposes a top-down framework to evaluate the importance of tokens for pruning. (Meng et al., 2022) selects and discards from the perspective of patches, tokens and blocks respectively. (Liang et al., 2022) calculates the attentiveness of the class token with respect to each image token and then the attentiveness value is used as criterion to identify the top-k attentive tokens. (Wang et al., 2022) compresses ViTs in frequency domain to prune both parameters and tokens in a unified framework. (Marin et al., 2021) designs a method to select cluster center which helps to do token pooling operations. (Fayyaz et al., 2022) scores and samples tokens according to their significance. (Kong et al., 2022) designs a learnable token selection module, which keeps the token with rich information and packs the token with less information into one and sends it to the next layer. Another methods (Bolya et al., 2023; Heo et al., 2023) focus on fusing similar tokens together. For instance, (Bolya et al., 2023) partitions tokens into two sets by random or odd/even partition and fuses each most similar token pairs between two sets.

# 3 METHOD

To reduce the computational cost, we propose to condense tokens along with forward process of vision transformers layer by layer. The token condensation contains two steps, $i.e.$, token selection and fusion. Different from directly discarding in token pruning method, we firstly select several most representative tokens and then separately fuse others into them.

Whether selecting representative tokens or fusing highly correlated ones, we both require to model global relationship between them. For this reason, we consider a factor, $Connectivity$, which measures how relevant a token is to another one. For instance, in a simplest case, we could use the attention value from one patch token to another one calculated from self-attention layer to describe a trivial connectivity. When one token is more relevant to others, it is easier to be replaced. When one token is less relevant to others, it is harder to be replaced. Furthermore, we consider the influence of other tokens when calculating the connectivity between two tokens, which we call as high-order connectivity. In order to help select representative tokens, we also consider another factor, $Dominance$, which measures the task relevance of tokens. For instance, in classification task, it could be describe as the attention values from class token to patch tokens. By using connectivity and dominance values, we could calculate a representative score. When the token's score is high, we tend to preserve it. Thus, ordered by descending way based on this representative score, the top tokens will be selected and preserved as representative ones and others will be selected as unrepresentative ones. In the fusion step, we use connectivity to match the unrepresentative tokens with the representative ones, and then we integrate the formers into the latters respectively. After condensation, new integrated tokens will be sent to next layer. Our method is inserted in each transformer layer between self-attention layer and MLP layer. The pipeline is shown in Figure 1.

Section 3.1 introduces how token connectivity and dominance are calculated. Section 3.2 introduces how tokens are selected and fused.

## 3.1 HIGH-ORDER CONNECTIVITY BETWEEN TOKENS

In vision transformers, the redundancy of tokens gives us room for optimization, $i.e.$ reducing the total token number by condensation. Because our method containing both selecting representative tokens and fusing highly correlated ones, we define the factor, $Connectivity$, noted as $\mathcal{C}$, to measure the relevance between tokens. $\mathcal{C}$ reflects both the direct and indirect relationships between all tokens. The direct relationship, for instance, could be describe as attention value calculated by one token's query and another token's key, which does not involve the addition of other tokens information to affect the result. The indirect relationship could be calculated by several direct relationships, for example, several attention values. For instance, we could firstly calculate the attention values of the two tokens for a third one. When we combine these two attention values together, for example by product, we could obtain the indirect relationship between these two token with information added from the third one. In this way, we could not only get the connectivity between two tokens directly but also allow this connectivity to be influenced by other tokens so as to obtain richer global information.

Here we will introduce how $\mathcal{C}$ is defined and derived. In order to describe the connectivity between tokens, we define a directed weighted graph $\mathcal{G} = (\mathcal{V}, \mathcal{E}, \mathcal{W})$, where $\mathcal{V}$ denotes the node set and $\mathcal{E}$ denotes the directed edge set. $\mathcal{W}$, as a matrix with size of $|\mathcal{V}| \times |\mathcal{V}|$, is the edge weight matrix of graph $\mathcal{G}$. For node $i$ and $j$: $\mathcal{V}(i)$ represents node $i$ in graph $\mathcal{G}$; $\mathcal{E}(i, j)$ represents the directed edge points from $\mathcal{V}(i)$ to $\mathcal{V}(j)$; $\mathcal{W}[i, j]$ represents the weight of $\mathcal{E}(i, j)$. For an image with size of $H \times W \times 3$, we divide it into $N^2$ patches with size of $\frac{H}{N} \times \frac{W}{N}$ as most transformer-based models did. The patches we got, also known as tokens, finally form the nodes set $\mathcal{V}$ of the graph $\mathcal{G}$. The direct relationship among all tokens will be measured by $\mathcal{W}$.

In order to obtain global information to condense tokens, we further introduce indirect relationship between every token pair. We add global information from other tokens by defining $k$-order connectivity $\mathcal{C}_k \in \mathcal{R}^{N^2 \times N^2}$, which captures the relationship among nodes in high-order. For example, in a node sequence, we could obtain the high-order relationships between the first and the last one by merging the direct relationship of all adjacent nodes. The higher the order is, the more the nodes will be involved in the calculation. In this way, more and more global information will be captured and whether a token is unique or not will be characterized more clearly. In this way, after we get all $\mathcal{C}_i$

with $i \in \{1, 2, 3...K\}$, we sum it up to finally obtain the connectivity $\mathcal{C}$, which contains all order relationships.

For example, we randomly pick up two tokens $i, j \in \mathcal{V}$. $\mathcal{C}_k[i,j]$ means $k^{th}$-order connectivity from $\mathcal{V}(i)$ to $\mathcal{V}(j)$. The first-order connectivity from $\mathcal{V}(i)$ to $\mathcal{V}(j)$ is measured by $\mathcal{W}[i,j]$, which reflects the information directly passing from $\mathcal{V}(i)$ to $\mathcal{V}(j)$. That is $\mathcal{C}_1 = \mathcal{W}$. The second-order connectivity from $\mathcal{V}(i)$ to $\mathcal{V}(j)$ is measured by the sum of information starting from $\mathcal{V}(i)$, passing through other intermediate nodes in one step, and finally reaching $\mathcal{V}(j)$ in another step. It could be described as

$$\mathcal{C}_2[i,j] = \alpha \sum_{m \in \mathcal{V}; \mathcal{E}(i,m), \mathcal{E}[m,j] \in \mathcal{E}} \mathcal{W}[i,m] \times \mathcal{W}[m,j] \tag{1}$$

where $\alpha$ is an adjustment factor. In this way, when we consider the relationship from $\mathcal{V}(i)$ to $\mathcal{V}(j)$, other nodes also contribute a lot and provide global information. Equation 1 could be written as a vector dot product

$$\mathcal{C}_2[i,j] = \alpha \mathcal{W}[i,:] \cdot \mathcal{W}[:,j] \tag{2}$$

where $\cdot$ denotes the vector dot product, $\mathcal{W}[i,:]$ denotes the $i^{th}$ row of $\mathcal{W}$ and $\mathcal{W}[:,j]$ denotes the $j^{th}$ column of $\mathcal{W}$. Thus, we could obtain $\mathcal{C}_2$ as

$$\mathcal{C}_2 = \alpha \mathcal{W}^2 \tag{3}$$

So in the same way as above, when we tend to calculate $\mathcal{C}_k[i,j]$ where $k > 2$, we should first calculate $\mathcal{C}_{k-1}[i,m]$, where m is a intermediate node $\mathcal{V}(m)$ different from $\mathcal{V}(i)$ and $\mathcal{V}(j)$. Then we calculate through a node sequence, starting from $\mathcal{V}(i)$, passing through $\mathcal{V}(m)$ and finally reaching $\mathcal{V}(j)$, by $\alpha \mathcal{C}_{k-1}[i,m] \times \mathcal{W}[m,j]$. Thus, in order to calculate $\mathcal{C}_k$, we need to sum up all the values like $\alpha \mathcal{C}_{k-1}[i,m] \times \mathcal{W}[m,j]$ with all different $m$. It could be written as

$$\mathcal{C}_k[i,j] = \alpha \sum_{m \in \mathcal{V}, \mathcal{E}[m,j] \in \mathcal{E}} \mathcal{C}_{k-1}[i,m] \times \mathcal{W}[m,j] \tag{4}$$

Same as previous derivation, equation 4 could also be written as a vector dot product

$$\mathcal{C}_k[i,j] = \alpha \mathcal{C}_{k-1}[i,:] \cdot \mathcal{W}[:,j] \tag{5}$$

Thus, we could obtain $\mathcal{C}_k$ as

$$\mathcal{C}_k = \alpha \mathcal{C}_{k-1} \mathcal{W} \tag{6}$$

Thus, combining equation 3, 6, we could obtain $\mathcal{C}_k$ as

$$\mathcal{C}_k = \alpha \mathcal{C}_{k-1} \mathcal{W} = \alpha^2 \mathcal{C}_{k-2} \mathcal{W}^2 = ...... = \alpha^{k-2} \mathcal{C}_2 \mathcal{W}^{k-2} = \alpha^{k-1} \mathcal{W}^k \tag{7}$$

According to the above, if we want to obtain both direct and indirect relationship among tokens to finally obtain the complete connectivity $\mathcal{C}$, we should sum up all the connectivity from first-order to $K^{th}$-order, where $K$ is a positive integer. Thus,

$$\mathcal{C} = \sum_{k=1}^{K} \mathcal{C}_k \tag{8}$$

where $\mathcal{C} \in \mathcal{R}^{N^2 \times N^2}$. The equation 8 can be expanded and derived as follows.

$$\begin{aligned}
\mathcal{C} &= \mathcal{W} + \alpha \mathcal{W}^2 + ... + \alpha^{K-1} \mathcal{W}^K \\
&= \mathcal{W}(1 + \alpha \mathcal{W} + ... + \alpha^{K-1} \mathcal{W}^{K-1}) \\
&= \mathcal{W}(1 + \alpha \mathcal{W} + ... + \alpha^{K-1} \mathcal{W}^{K-1})(1 - \alpha \mathcal{W})(1 - \alpha \mathcal{W})^{-1} \\
&= \mathcal{W}(1 - \alpha^K \mathcal{W}^K)(1 - \alpha \mathcal{W})^{-1}
\end{aligned} \tag{9}$$

Here $(1 - \alpha \mathcal{W})^{-1}$ means the inverse of matrix $(1 - \alpha \mathcal{W})$. Considering that solving the inverse of the matrix has a high computational cost, and the inverse of the matrix is difficult to do gradient backward after the calculation is completed, we choose to use Newton iterative method (Ogden, 1969; Guo & Higham, 2006) to approximate the matrix inversion. For convenience, we denote $(1 - \alpha \mathcal{W})$ as $\mathcal{A}$. We assume that the result of the iterative method at step $n$ is $X_n$. According to (Ogden, 1969; Guo & Higham, 2006), the iterative process of calculating the matrix $\mathcal{A}$'s inverse is achieved by the formula following

$$X_{n+1} = X_n + X_n(I - \mathcal{A} X_n) \tag{10}$$

According to (Ogden, 1969; Guo & Higham, 2006), if $X_n$ is defined as in equation 10 for $n = 1, 2, ...$, then $\mathcal{A}^{-1} = \lim_{n \to +\infty} X_n$ if and only if $||I - \mathcal{A}X_0|| < 1$.

In actual implementation, we assign $\mathcal{W}$ to attention calculated by matrix query and key in transformer and additionally conduct ablation experiments with other metrics in section 4.2. Thus, the absolute values of diagnose of $\mathcal{W}$ are always bigger than others and mostly $\mathcal{W}$ is a diagonally dominant matrix. We set $\alpha$ as a relatively large value which helps that, after calculation of $\mathcal{A} = I - \alpha\mathcal{W}$, $\mathcal{A}$ is still a diagonally dominant matrix, which is relatively easy to satisfy the condition of $||I - \mathcal{A}X_0|| < 1$. In order to facilitate the calculation and meet the convergence conditions, we assign $X_0$ to the reciprocal of the diagnose of matrix $\mathcal{A}$ and set the number of iteration steps to less than three.

Here we finally get the connectivity value $\mathcal{C}$ token to token. To get each token's connectivity from the whole set $\mathcal{V}$, we do mean operation to $\mathcal{C}$ on token dimension. Thus, we finally get the connectivity of each token $\bar{\mathcal{C}} \in \mathcal{R}^{1 \times N^2}$.

Besides, the $Dominance$ is noted as $\mathcal{D} \in \mathcal{R}^{1 \times N^2}$. Each value in $\mathcal{D}$ reflects the relationship from image token to class token. The $i^{th}$ value in $\mathcal{D}$, noted as $\mathcal{D}[i]$, is calculated by self-attention from the query of class token to the key of patch $i$'s token in actual implementation and other metrics results are listed in section 4.2. After the calculation, we naturally use a softmax calculation to normalize the calculated value between 0 and 1.

Combining $\mathcal{C}$ and $\mathcal{D}$, we finally get the representative score function. According to statement above, when a token's connectivity is small, it is difficult to be replaced by other tokens. Thus, the lower the connectivity between a token and others is, the stronger its unique representation will be. It means that connectivity is negatively correlated with token's representativeness. Besides, when a token is more relevant to the task, it is more representative. It means dominance is positively correlated to token's representativeness. Based on the above, we could finally define the selection score function as

$$\mathcal{S} = -\bar{\mathcal{C}} + \beta\mathcal{D} \tag{11}$$

where $\mathcal{S} \in \mathcal{R}^{1 \times N^2}$ and $\beta$ is a scale factor to balance $\bar{\mathcal{C}}$ and $\mathcal{D}$. Each value in $\mathcal{S}$ reflect the representativeness for its corresponding token.

## 3.2 SELECT AND FUSE

**Select.** After calculating the representative score $\mathcal{S}$ of each token, we could sort these tokens from large to small. Because the time cost of each module of our method is not high, we could insert it into every transformer block between attention layer and mlp layer. Thus, it provides a lot of flexibility to choose the number of reserved tokens for each layer. At layer $i$, suppose the input number of tokens is $N_i^{in}$ and output one is $N_i^{out}$. The preserved tokens form the set $\mathcal{P}_i = \{t_{p_1}, t_{p_2}, ..., t_{p_{N_i^{out}}}\}$ and other tokens form the set $\mathcal{Q}_i = \{t_{q_1}, t_{q_2}, ..., t_{q_{(N_i^{in} - N_i^{out})}}\}$.

**Fuse.** Our target is to fuse all the tokens in the discarding set $\mathcal{Q}_i$ into tokens in the preserving set $\mathcal{P}_i$. For each token in $\mathcal{Q}_i$, we firstly find the token that best matches it in $\mathcal{P}_i$. In actual implementation, we use connectivity as the matching criterion. In this way, for each $t_q \in \mathcal{Q}_i$, we could get the most connected token $t_p \in \mathcal{P}_i$ with it. All tokens like $t_p$ above form a set $\mathcal{T}_i \subseteq \mathcal{P}_i$. Then for each token $t$ in $\mathcal{P}_i$ to be retained, $i.e.$ $t \in \mathcal{T}_i$, we could obtain 0 or several tokens discarded in set $\mathcal{Q}_i$ best match to it, which form a set noted as $\mathcal{T}_t'$. In this way, it is equivalent to use one token to replace the roles of several associated tokens. These best matched tokens will be fused into the representative token in $\mathcal{P}_i$ by connectivity-based weighted averaging operation. Thus, The token $t$ will be fused as

$$t = \frac{t\mathcal{C}[t,t] + \sum_{t' \in \mathcal{T}'} t'\mathcal{C}[t',t]}{\mathcal{C}[t,t] + \sum_{t' \in \mathcal{T}'} \mathcal{C}[t',t]} \tag{12}$$

## 4 EXPERIMENTS

In this section, we empirically investigate the effectiveness of our proposed token condensation method for efficient vision transformer (TC-ViT). We evaluate our method on the benchmark ImageNet(ILSVRC2012) (Deng et al., 2009) dataset, which contains 1000-class natural images, including 1.2M training images and 5k validation images. We firstly compare the proposed method with other SOTA token pruning methods. Then we conduct ablation studies to better understand our methods.

## 4.1 Experiments on ImageNet

**Implementation Details.**    We conduct experiments on different backbones including DeiT-Tiny, DeiT-Small, DeiT-Base (Touvron et al., 2020) and LV-ViT-S, LV-ViT-M (Jiang et al., 2021). By default, we set K to 2, $\alpha$ to 500 and $\beta$ to 3.5. According to (Tang et al., 2022), the way of token similarity increasing in depth is approximately linear growth. (Bolya et al., 2023) found that the linear strategy for token pruning layer by layer is the best. Thus we follow (Tang et al., 2022; Bolya et al., 2023) and set the number of discarded tokens in each layer as a constant. We follow the training and testing settings in the original papers (Touvron et al., 2020; Jiang et al., 2021) and our method is implemented based on the official pretrained models. After inserting our modules, the model after pruning is finetuned following the training strategy in (Touvron et al., 2020; Rao et al., 2021; Tang et al., 2022), where the learning rate is set to 1e-4 and distillation weight is set to 0.5 (Yin et al., 2022; Kong et al., 2022). All the experiments conducted with PyTorch (Paszke et al., 2017) on 8 NVIDIA V100 GPUs.

**Main Results.**    The experiment results are shown in Table 1. We compare our method with existing token pruning methods (Rao et al., 2021; Yin et al., 2022; Kong et al., 2022; Xu et al., 2022; Tang et al., 2022; Dong et al., 2023; Liang et al., 2022; Bolya et al., 2023; Fayyaz et al., 2022) on basic model DeiT. Most methods mentioned above, such as (Rao et al., 2021; Yin et al., 2022; Kong et al., 2022; Xu et al., 2022; Tang et al., 2022; Dong et al., 2023; Liang et al., 2022), apply distillation in their final criterion. For a fair comparison, we report the top-1 accuracy with distillation as well as GFLOPs and throughput for performance evaluation. Results without distillation will be reported in section 4.2. The throughput is measure on a single NVIDIA V100 GPU with batch size fixed to 128 and image size of $224 \times 224$, which are same as the settings of (Touvron et al., 2020; Tang et al., 2022). Our method achieves obviously higher performance compared to the existing methods. For instance, on DeiT-T and DeiT-S, we can seperately reduce the FLOPs by 46% and 50% without any accuracy decrease; on DeiT-B, we reduce the FLOPs by 50.6% and only have a small accuracy decrease(0.2%). We also obtain very competitive throughput results that the DeiT-T/S/B model achieves 48.0%/88.9%/93.1% speedup after token condensation.

We then further adopt our method on LV-ViT (Jiang et al., 2021). To simplify, the token labels in the original LV-ViT models are not used during fine-tuning and we do not use distillation, following (Tang et al., 2022). The results are shown in Table 2. Compared with (Tang et al., 2022), we reduce the FLOPs and increase the accuracy.

## 4.2 Ablation Study

We conduct extensive ablation studies on ImageNet to verify the effectiveness of each component in our method. The DeiT-T model on the ImageNet dataset is used as the basic model.

**Connectivity and Dominance.**    In order to verify the effectiveness of our method, we performed an ablation experiment on $\mathcal{C}$ and $\mathcal{D}$. We separately define score function as $-\mathcal{C}$, $\mathcal{D}$ and $-\mathcal{C} + \beta\mathcal{D}$. We set the number of discarded tokens in each layer as 14. The results are shown in Table 3. We could see that when we just use the connectivity to select representative tokens, we could only obtain the accuracy of 70.4%. When we just use the dominance to select, we could only obtain the accuracy of 71.9%. When we use both connectivity and dominance to select, we could obtain the accuracy of 72.3%. In this way, we verify that both connectivity and dominance contribute greatly to the results, and both are indispensable, which verify the effectiveness of our method.

**Fuse or not.**    In order to verify the validity of the fusion, we conduct experiment on our strategy w/ and w/o fusion. We also set the number of discarded tokens in each layer as 14. Results are shown in Table 4. Without the operation of fusion, the accuracy will drop 0.2%. We could see that it is important to fuse unpreserved token into preserved ones so as to guarantee the information integrity.

**Different metrics.**    In order to verify the effectiveness of attention score, we separately conduct experiments on the initialization of $\mathcal{W}$ and $\mathcal{D}$ in equation 9. $\mathcal{W}$ is successively defined as attention, cosine similarity, dot and Euclidean distance among image tokens and the results are shown in table 5. From the result, we could find that $attn$ achieves higher performance because it can reflect the tokens' partial order relationship more reasonably and it is more in line with token aggregation process.

Table 1: Main results on ImageNet compared with other existing token pruning methods on DeiTs. 'FLOPs ↓' denotes the reduction ratio of FLOPs. 'Throughput ↑' denote the growth ratio of Throughput. Following (Tang et al., 2022), all the throughputs are tested with batch size of 128 on a single V100 GPU with PyTorch.

| Model | Method | Top-1 Acc. (%) | FLOPs (G) | FLOPs ↓ (%) | Throughput (img/s) | Throughput ↑ (%) |
|-------|--------|----------------|-----------|-------------|--------------------|--------------------|
| DeiT-T | Baseline (Touvron et al., 2020) | 72.2 | 1.3 | 0 | 2.5K | 0 |
| | A-ViT (Yin et al., 2022) | 71.3 | 0.8 | 38.4 | 3.4K | 36.0 |
| | ToMe (Bolya et al., 2023) | 71.4 | 0.8 | 38.4 | - | - |
| | PS-ViT (Tang et al., 2022) | 72.0 | 0.7 | 46.2 | 3.6K | 44.0 |
| | Evo-ViT (Xu et al., 2022) | 72.0 | - | - | 3.9K | 57.6 |
| | HeatViT (Dong et al., 2023) | 72.1 | 0.9 | 30.8 | - | - |
| | SPViT (Kong et al., 2022) | 72.1 | 0.8 | 38.4 | - | - |
| | **TC-ViT(ours)** | **72.3** | **0.7** | **46.2** | **3.7K** | **48.0** |
| DeiT-S | Baseline (Touvron et al., 2020) | 79.8 | 4.6 | 0 | 0.9K | 0 |
| | A-ViT (Yin et al., 2022) | 78.9 | 3.6 | 21.7 | 1.1K | 22.2 |
| | ToMe (Bolya et al., 2023) | 79.2 | 2.4 | 47.8 | - | - |
| | DynamicViT (Rao et al., 2021) | 79.3 | 2.9 | 37.0 | - | - |
| | HeatViT (Dong et al., 2023) | 79.3 | 2.6 | 43.5 | - | - |
| | SPViT (Kong et al., 2022) | 79.3 | 2.6 | 43.5 | - | - |
| | Evo-ViT (Xu et al., 2022) | 79.4 | - | - | 1.4K | 66.7 |
| | EViT (Liang et al., 2022) | 79.5 | 3.0 | 34.8 | - | - |
| | PS-ViT (Tang et al., 2022) | 79.5 | 2.4 | 47.8 | 1.3K | 44.4 |
| | ATS (Fayyaz et al., 2022) | 79.7 | 2.9 | 37.0 | 1.5K | 55.6 |
| | **TC-ViT(ours)** | **79.9** | **2.3** | **50.0** | **1.7K** | **88.9** |
| DeiT-B | Baseline (Touvron et al., 2020) | 81.8 | 17.6 | 0 | 0.29K | 0 |
| | EViT (Liang et al., 2022) | 81.3 | 11.6 | 34.1 | - | - |
| | DynamicViT (Rao et al., 2021) | 81.3 | 11.2 | 36.4 | - | - |
| | Evo-ViT (Xu et al., 2022) | 81.3 | - | - | 0.44K | 58.7 |
| | PS-ViT (Tang et al., 2022) | 81.5 | 9.8 | 44.3 | 0.41K | 41.4 |
| | **TC-ViT(ours)** | **81.6** | **8.7** | **50.6** | **0.56K** | **93.1** |

Table 2: Main results for LV-ViT on ImageNet w/o distillation.

| Model | Method | Top-1 Acc. (%) | FLOPs (G) | FLOPs ↓ (%) | Throughput (img/s) | Throughput ↑ (%) |
|-------|--------|----------------|-----------|-------------|--------------------|--------------------|
| LV-ViT-S | Baseline (Jiang et al., 2021) | 83.3 | 6.6 | 0 | 0.71K | 0 |
| | PS-ViT (Tang et al., 2022) | 82.4 | 4.7 | 28.8 | 0.90K | 26.8 |
| | **TC-ViT(ours)** | **82.5** | **4.1** | **37.9** | **0.99K** | **39.4** |
| LV-ViT-M | Baseline (Jiang et al., 2021) | 84.0 | 12.7 | 0 | 0.37K | 0 |
| | PS-ViT (Tang et al., 2022) | 83.5 | 8.6 | 32.3 | 0.59K | 59.4 |
| | **TC-ViT(ours)** | **83.5** | **7.4** | **41.7** | **0.68K** | **83.8** |

**Distillation.** In order to further obtain more effective results, we also conduct experiments without distillation. The results are shown in table 6. The results show that our method can still significantly accelerate inference with controllable accuracy drop.

Table 3: Comparation on different score functions w/ or w/o connectivity and dominance.

| Connectivity | Dominance | Top-1 Acc.(%) |
|--------------|-----------|---------------|
| ✓ | ✗ | 70.4(-1.9) |
| ✗ | ✓ | 71.9(-0.4) |
| ✓ | ✓ | 72.3 |

Table 4: Comparation on whether using fusion strategy.

| Fusion | Top-1 Acc.(%) |
|--------|---------------|
| ✗ | 72.1(-0.2) |
| ✓ | 72.3 |

Table 5: Comparison on different $\mathcal{W}$ and $\mathcal{D}$ initialization methods.

| Initialization on $\mathcal{W}$ and $\mathcal{D}$ | attn | cos | dot | eucl |
|---|---|---|---|---|
| Top-1 Acc.(%) | 72.3 | 72.1 | 71.9 | 72.0 |

Table 6: Main results w/o distillation compared with other existing methods on DeiTs.

| Model | Method | Top-1 Acc.(%) | FLOPs (G) | Throughput (img/s) |
|---|---|---|---|---|
| DeiT-T | A-ViT (Yin et al., 2022) | 71.0 | 0.8 | 3.4K |
| | **TC-ViT(ours)** | **71.5** | **0.8** | **3.4K** |
| DeiT-S | A-ViT (Yin et al., 2022) | 78.6 | 3.6 | 1.1K |
| | DynamicViT (Rao et al., 2021) | 79.2 | 2.9 | - |
| | **TC-ViT(ours)** | **79.3** | **2.3** | **1.7K** |

**Balance between accuracy and efficiency.** To measure the performance and efficiency of our method, we conduct experiments with different number of discarded tokens in each layer and the results are shown in Table 7. We also draw a curve of relation between accuracy and GFLOPs, which is shown in Figure 2. By increasing the number of discarded tokens in each layer, more tokens can be pruned to achieve higher acceleration rate. When the reduction of FLOPs is less than 45%, we could obtain results without any loss of precision, or even more precise, which is because we not only prune the number of token to reduce the computation cost but also preserve most information through fusion method.

**Connectivity order.** In order to verify the effectiveness between connectivity order and efficiency, we conduct experiments with different K. As same before we set dropped number as 14. We show the result in Table 8. We could find that if the K value is too low, there will be a certain degree of precision loss and although a larger K value can still achieve competitive result to a certain extent, it will bring a higher calculation cost. Therefore we compromise K to be 2 as our default setting to balance the accuracy and efficiency.

## 4.3 VISUALIZATION

**Diagonally Dominant Matrix.** In section 3.1, when we use Newton method to approximate the result of $(1 - \alpha\mathcal{W})^{-1}$, we claim $(1 - \alpha\mathcal{W})$ approximately to a diagonally dominant matrix. We visualize first six layers of $|1 - \alpha\mathcal{W}|$ in Figure 3 for a image in ImageNet validation set. We could find that in most instance, the neighbor of diagnose in figures are more lighter than other regions, which means $(1 - \alpha\mathcal{W})$ mostly is a diagonally dominant matrix.

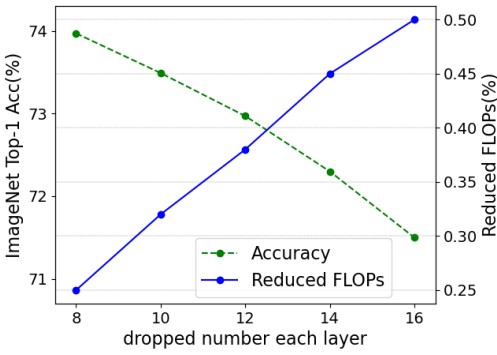

Figure 2: The ImageNet accuracy and FLOPs of condensed DeiT-T w.r.t different numbers of discarded tokens in each layer.

Table 7: Different numbers of discarded tokens in each layer on Deit-T .

| Model | Drop number | Top-1 Acc.(%) | GFLOPs |
|-------|-------------|---------------|--------|
|       | 8           | 73.9          | 0.97   |
|       | 10          | 73.5          | 0.89   |
| DeiT-T | 12         | 72.9          | 0.81   |
|       | 14          | 72.3          | 0.73   |
|       | 16          | 71.5          | 0.65   |

Table 8: Results of different order K on DeiT-T.

| K  | GFLOPs | Top-1 Acc. (%) |
|----|--------|----------------|
| 1  | 0.69   | 72.0           |
| 2  | 0.73   | 72.3           |
| 4  | 0.76   | 70.2           |
| 8  | 0.79   | 71.9           |
| 32 | 0.85   | 72.0           |

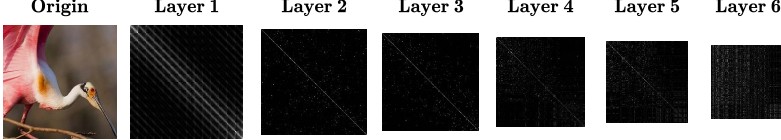

Figure 3: Visualizations of $|1 - \alpha \mathcal{W}|$ in first six layers.

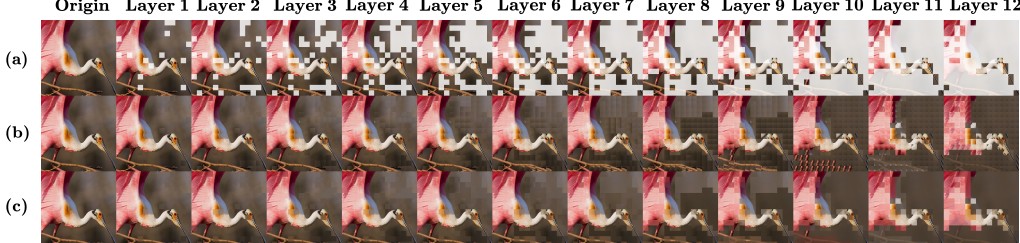

Figure 4: Visualizations of each layer output. Row (a) visualizes the tokens retained in each layer. Row (b) visualizes how discarded positions are directly replaced by preserved tokens. Row (c) visualizes the results after we fuse all matched tokens.

**Token Condensation.**    We visualize some results of pictures in ImageNet validation set on DeiT-T, dropping 14 tokens in each layer. It is shown in Figure 4 as examples. The figure contains three rows and thirteen columns. In these thirteen columns, from left to right, there are the original image, and the output results of the first to twelfth layer. In these three rows, row (a) is the visualization result of the tokens retained in each layer. The white positions are masked to represent those discarding ones. We could find that our method not only figures out which part of the image is representative but also preserves some part of the background to help describe the whole picture with less tokens. Row (b) is the visualization result of directly filling the discarded token positions with the reserved ones as substitution. We could clearly observed that how those redundant tokens are fused into representative ones step by step. Some tokens, once still representative in the former layer, may become relatively redundant in the next layer. Row (c) is the visualization of mean of all matched tokens. Once several tokens are fused into a new token, this new token will replace all original tokens positions. We could find that not only our method could pick out representative tokens, but also when using these representative tokens to fill in the lost parts, the reconstructed image still has strong semantic information. More visualization results are shown in supplemental materials.

## 5    CONCLUSION

To reduce token redundancy, we propose a new token condensation method by selecting and fusing tokens. The metric connectivity is proposed to help select representative tokens and match relevant tokens before fusion. The fusion of tokens can retain all the information to the greatest extent. As our method is light-weighted, we could insert it at each layer and prune tokens smoothly. Extensive experiments on benchmark datasets validate that the proposed method can effectively reduce the computation cost.

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

# A  APPENDIX

## A.1  COMPARISON ON DIFFERENT HYPER PARAMETER

We further conduct experiments on hyper parameter $\alpha$ in calculation of connectivity $\mathcal{C} = \mathcal{W}(1 - \alpha^K \mathcal{W}^K)(1 - \alpha \mathcal{W})^{-1}$ and $\beta$ in calculation of score function $\mathcal{S} = -\bar{\mathcal{C}} + \beta \mathcal{D}$. We use DeiT-Tiny as basic model.

For experiments on $\alpha$, we set K as 2, $\beta$ as 3.5, and number of discarded tokens in each layer as 14. The results are shown in Table 9. When $\alpha$ is 0, it is equal to the method without high-order connectivity. We could find that a larger value of $\alpha$ will achieve better results to some extend because a larger $\alpha$ leads to more attention on high-order connectivity.

Table 9: Comparison with different $\alpha$.

| $\alpha$ | Top-1 Acc. (%) |
|---|---|
| 1000 | 72.2 |
| 600 | 72.2 |
| 500 | 72.3 |
| 400 | 72.3 |
| 200 | 71.9 |
| 0 | 72.0 |

For experiments on $\beta$, we set K as 2, $\alpha$ as 500, and number of discarded tokens in each layer as 14. The results are shown in Table 10. We could find that when $\beta$ is too small, it will degenerate into the method only using connectivity and when $\beta$ is too large, it will degenerate into the method only using dominance. Both of the circumstances lead to the performance degradation.

Table 10: Comparison with different $\beta$.

| $\beta$ | Top-1 Acc. (%) |
|---|---|
| $+\infty$[1] | 71.9 |
| 5.0 | 72.1 |
| 4.0 | 72.3 |
| 3.0 | 72.3 |
| 2.0 | 72.2 |
| 1.0 | 72.1 |
| 0 | 70.4 |

## A.2  VISUALIZATION ON IMAGENET VALIDATION SET

Here we visualize more results to show our method in Figure 5. We present our results group by group. In each group, we still present it in three rows and thirteen columns. In row (a), there are the results of how representative tokens are selected in each layer. We could find that out method mostly focuses on objects while still retains attention to other information, $i.e.$ background, to some extend. It not only highlights the main target, but also preserves the complete semantic information of the image as far as possible. In row (b), there are the results of images reconstructed by those representative tokens in each layer, which do not use the fusion strategy. In row (c), there are the results of images reconstructed by tokens after fusion step in each layer. We could find that sometimes when the number of discarded tokens becomes larger, without fusion step, the reconstructed images look even more unreal. On the other hand, when tokens are fused in each layer, the reconstructed images look more real with more complete semantic information.

---

[1]The $+\infty$ here means when $\beta$ is so large that we could ignore the influence of connectivity, which degenerates into the method only using dominance.

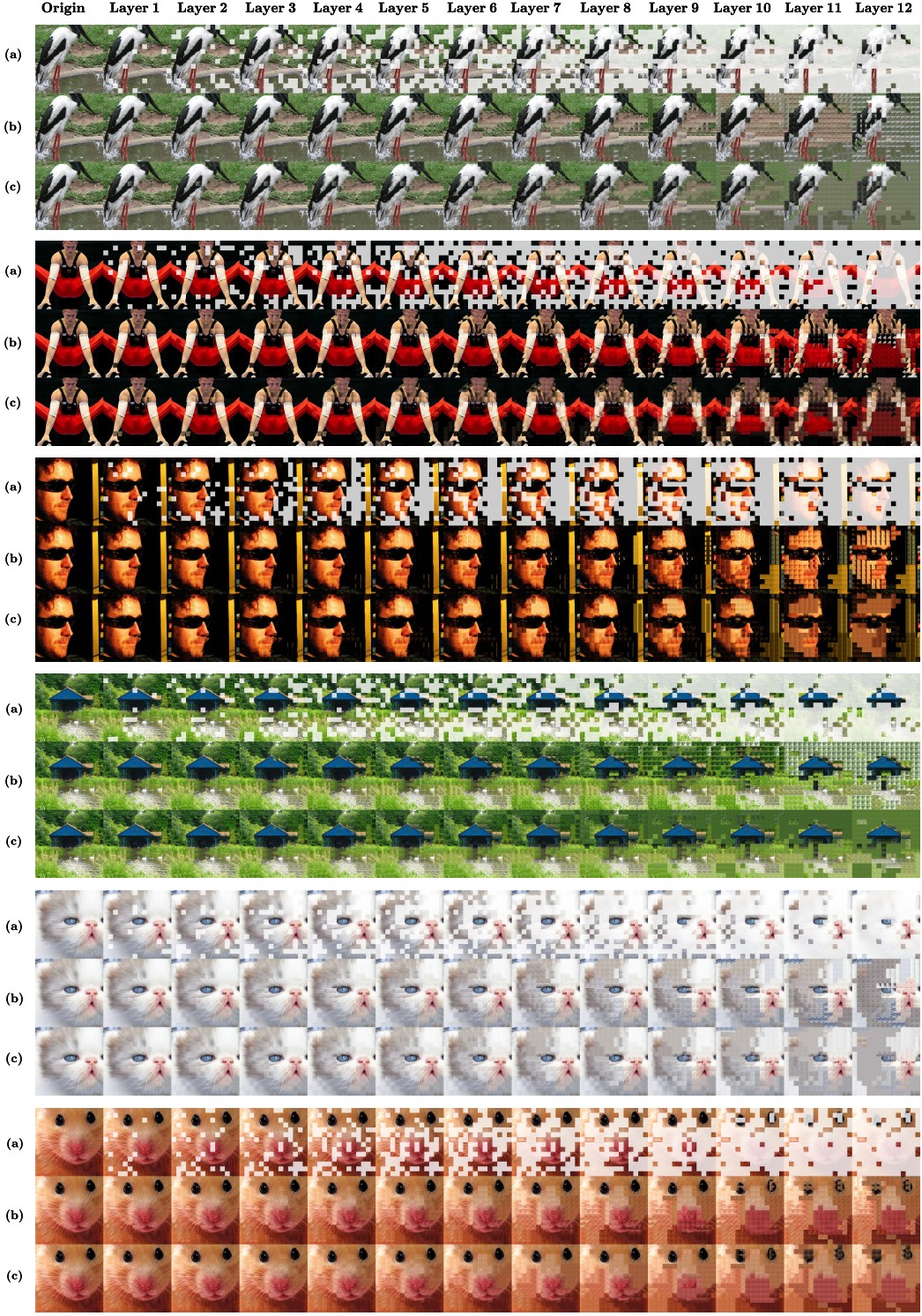

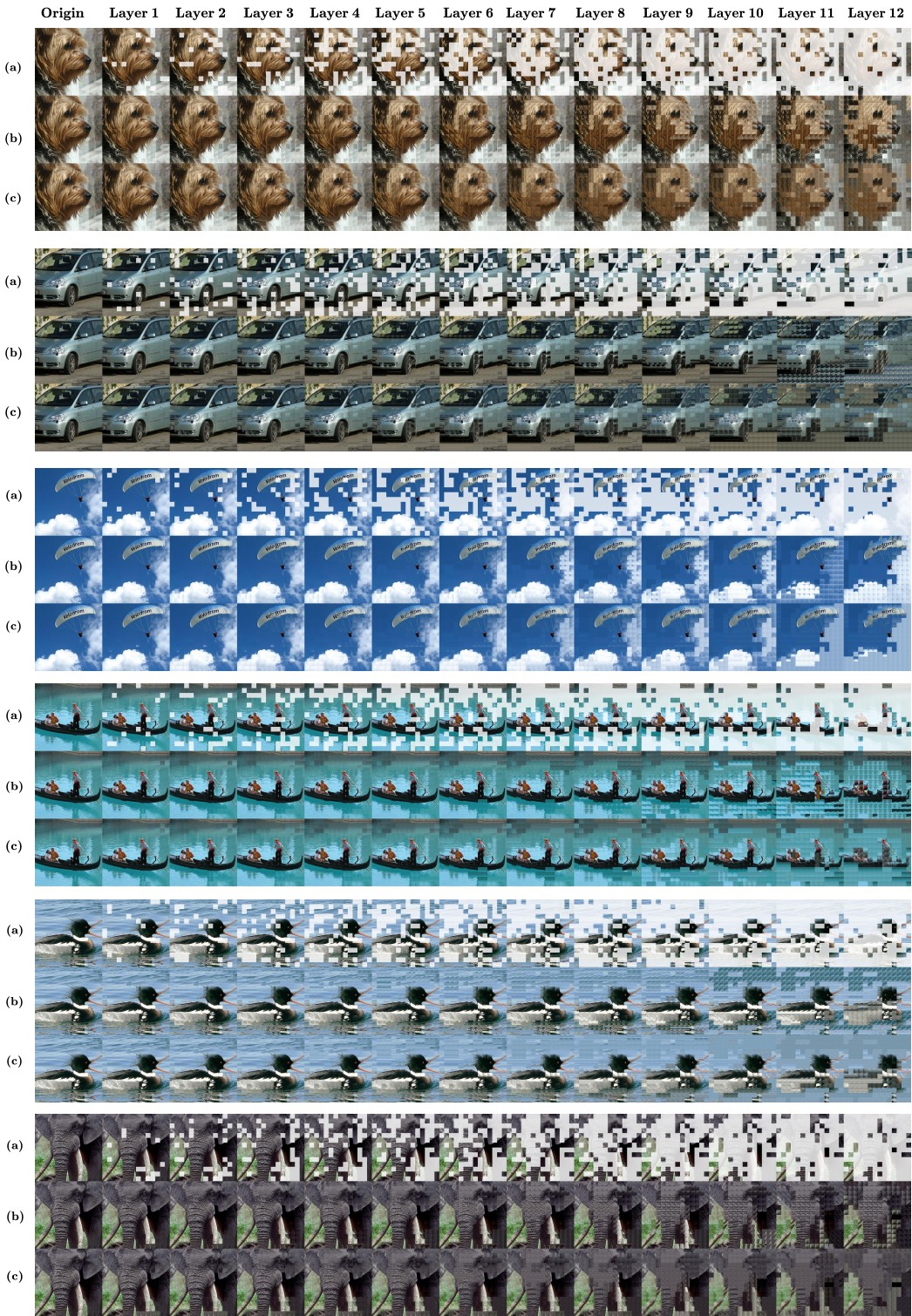

Figure 5: Visualizations of different layers. Images in row (a) are the results of how representative tokens are selected in each layer. Images in row (b) are the results reconstructed by those representative tokens in each layer without fusion strategy. Images in row (c) are the results reconstructed by tokens after fusion step in each layer.

