# OpenReview forum: "Connectivity-based Token Condensation for Efficient Vision Transformer"
_ICLR.cc/2024/Conference — ICLR 2024 Conference Withdrawn Submission_

### Official Review · Reviewer_V3W2 · 2023-10-30

**Soundness:** 2 fair
**Presentation:** 3 good
**Contribution:** 2 fair
**Rating:** 3
**Confidence:** 4

**Summary:**

The paper proposes a new method for **token pruning** in vision transformers, called `TC-ViT`. Most token pruning methods discard tokens based on their attention score relatively to the class token.

In contrast, `TC-ViT` also considers higher order connectivity between tokens: First, a fully-connected graph between tokens is considered. The first-level connectivity scores are obtained from the attention matrix $W$. The second-level connectivity scores $C_2(i, j)$ are obtained with a message-passing like process: essentially summing attention score along all the paths of length 2 from node $i$ to $j$. And this process can be iterated to obtain higher-level connectivity scores.
A second score, the dominance $D$ is obtained by considering the attention-score of each token to the class token, similarly to other token pruning methods. The final "relevance" score of each token is obtained as a weighted sum of both the connectivity and dominance values.
Then, in each layers, the tokens are split into two groups (highly relevant set $P$ and  not relevant ones $Q$); All irrelevant tokens in $Q$ are matched to their closest tokens in $P$, which are in turn updated/fused with the relevant tokens in $Q$. After being fused, the irrelevant tokens are discarded.

Finally, the proposed method is evaluated on the task of image classification for two backbones (`DeiT`-based and `LV-ViT/S`) and compared to several token pruning baselines. In addition, the paper considers several ablation experiments such as comparing the impact of the connectivity and dominance scores, whether to use the fusion strategy etc.

**Strengths:**

- **Thorough ablation experiments:** The paper reports several ablation experiments on different components of the proposed methods such as the connectivity level or the number of tokens dropped in the architecture.
- **Clear explanation of the method:** The method section clearly introduces and formalizes the proposed connectivity score as well as how to compute it by iteratively building higher-order connectivity information.

**Weaknesses:**

- **Unclear motivation**: Based on the introduction and results, it is not clear to me which is the main gap that the paper tries to adress:
  - For instance, when comparing to other token pruning methods, the introduction states that "*only discarding tokens will incur performance loss as even those less important tokens*" -> however, one could argue that merging tokens may also result in loss of information.  Furthermore, there are already existing methods such as "*EViT: Expediting vision transformers via token reorganizations, Liang et al, ICLR 2022*" which proposes to fuse all pruned tokens into one token, rather than discarding them.
  - Similarly, about token merging methods, the introduction mention that "*They also overlook the potential influence of other tokens on the
relationships between two tokens when analyzing their interactions*" -> However, when looking at the ablation experiments in **Table 8**, it seems that using a connectivity factor of $K = 1$ actually yields better performance than most of the higher connectivity values $K \in \{4, 8, 32\}$. This would indicate that higher-order relationships between tokens are not so meaningful for token pruning. Similarly, from the ablation in **Table 3**, it seems that the *dominance* score (i.e. the same attention to class token used in other token pruning methods) has a more significant impact on accuracy than *connectivity*.


- **Missing training hyperparmeters**: The experimental section is not very clear on the training setup of baselines. In particular, on whether **(i)** all baselines are trained from scratch or finetuned, **(ii)** how long they are trained for and **(iii)** whether they include distillation or not. Because of this and the fact that the accuracy gap is generally small, it is hard to interpret the results from **Table 1**; Furthermore, many results seem to directly come from their respective original paper, which might have a different training pipeline;  To give some examples of potential different training pipelines:
  - For DeiT-S The results reported for `EviT` (79.5% accuracy at 3.9GFlops) corresponds to the version trained from scratch for 300 epochs without distillation. If finetuned from a pretrained model, the original work reports 79.8% accuracy. Similarly for `ToME`, the results seems to correspond to training from scratch from the original paper.
  - According to the original paper, `HeatViT` also applies 8-bit quantization to weights/activations after token pruning, so the accuracy might not be directly comparable.

**Questions:**

- **Making the fuse operation discrete:** In Section 3.2, the fusing operation is described as a discrete matching, where all irrelevant tokens are matched to its most relevant preserved tokens and fused. Since the goal is to maximally preserve the information container in the discarded tokens, would it be possible to fuse all irrelevant tokens into all preserved tokens, weighted by their connectivity score ?

- **Cost of the select/fuse operation**: It would be interesting to report a more precise breakdown of the compute cost in terms of **(i)** computational cost from running the transformer versus **(ii)** computational cost from computing the connectivity score and selecting/fusing tokens.

- **Minor suggestion on writing:** The use of *could* in the method section is a bit confusing, as it is not clear if the paper describes something that the authors tried/considered, or something that is actually part of the final proposed method. For instance, in Section 3.2 "*we could sort these tokens from large to small. Because the time cost of each module of our method is not high, we could insert it into every transformer block between attention layer and mlp layer*

---

### Official Review · Reviewer_1bHr · 2023-11-01

**Soundness:** 3 good
**Presentation:** 2 fair
**Contribution:** 2 fair
**Rating:** 6
**Confidence:** 4

**Summary:**

This paper presents a connectivity-based token condensation method for efficient inference of vision Transformers. The paper introduces connectivity and dominance to model the relationship among tokens and uses these metrics to select tokens for fusing. The method is evaluated on ImageNet with several different backbones and compared to various existing methods.

**Strengths:**

- The method achieves good performance on ImageNet. It is impressive to see that the model acceleration method can outperform baseline models while achieving decent real speedup on GPUs.

- The paper overall is easy to read. It is reasonable to consider high-order relations to more accurately select representative tokens.

**Weaknesses:**

- The authors conducted several ablation experiments to validate the design choices mentioned in Section 3, but there are no direct comparisons between different token fusion strategies. As mentioned in the abstract, the authors claim that previous methods "do not consider carefully how to select tokens". To support this claim, I think it is necessary to provide direct evidence instead of the final performance of the whole system to show the superiority of the proposed designs. For example, ToMe (Bolya et al., 2023) presents a token fusion solution with bipartite matching. The method can achieve 79.3% with 2.7 GFLOPs without training based on AugReg ViT-S (Table 4). If the method can better select and fuse tokens, will the proposed method achieve better performance? Since obtaining the results doesn't require training, the comparisons can also reduce the effects of training techniques and settings.

- Why are there some missing numbers of FLOPs or throughput in Table 1 and 6? Are these results better than the proposed method? The code and models of most of these methods like ToMe and DynamicViT are publically available and widely used in previous work. I think it is easy to obtain the throughput of these models. It is also strange that you cannot obtain FLOPs of Evo-ViT while can measure the throughput of this model.

- Several recent token fusion/pruning methods [r1,r2,r3,r4] have extended this group of methods to more complex scenarios like convolution backbones and dense prediction tasks. Only conducting experiments on ImageNet based on DeiT and LVViT may make the work less competitive.

[r1] Joint Token Pruning and Squeezing Towards More Aggressive Compression of Vision Transformers, CVPR 23
[r2] ToMe (Bolya et al., 2023)
[r3] Dynamic Spatial Sparsification for Efficient Vision Transformers and Convolutional Neural Networks, TPAMI 23
[r4] Dynamic Token Pruning in Plain Vision Transformers for Semantic Segmentation ICCV 23

**Questions:**

The method is well-motivated and achieves good performance on widely used benchmarks. But there are several above-mentioned issues that require further discussion and clarification.  Therefore, I would like to rate this paper as  "marginally above the acceptance threshold". For questions, please refer to my comments above.

---

### Official Review · Reviewer_DqFk · 2023-11-01

**Soundness:** 2 fair
**Presentation:** 2 fair
**Contribution:** 2 fair
**Rating:** 3
**Confidence:** 4

**Summary:**

The paper proposes a method to make Vision Transformers (ViT) more efficient using a selection and fusion framework. In the selection process, tokens are divided into representative and unrepresentative sets according to certain sorted metrics. For fusion, the unrepresentative tokens are fused into representative ones. The most novel contribution appears to be the CONNECTIVITY-BASED metric used in selection and matching, which has the potential to capture high-order relationships between tokens.

**Strengths:**

The framework is clean and straightforward.
The performance is promising, as demonstrated by the DeiT-S results.

**Weaknesses:**

- The writing should be more concise.
- The novelty may not be sufficient for this conference. Although the framework has been proposed earlier, the main novelty comes from the CONNECTIVITY-BASED metric. A more detailed analysis of this metric should be provided, and a stronger explanation for how it captures "high-order relationships between tokens" is needed.
- In Equation 2, the dot product convention is denoted by \cdot, making further explanation redundant.
- The related work section is loosely written. The author should include several efficient ViT works following the same "selection then fusion" framework (e.g., [1]). The significant differences between these approaches should be highlighted.
- There is a lack of justification for the generalization of connection-based metrics for tasks other than classification.

[1] https://arxiv.org/abs/2304.10716

**Questions:**

- The results in Table 8 are confusing; could the authors explain the trend for accuracy for different orders? Why is the performance of K=4 significantly worse than the naive K=1 case?
- Can this approach be applied to tasks such as segmentation, besides classification?

---

### Official Review · Reviewer_apVU · 2023-11-02

**Soundness:** 2 fair
**Presentation:** 1 poor
**Contribution:** 2 fair
**Rating:** 5
**Confidence:** 4

**Summary:**

This work has proposed a new token pruning method for ViT in a selecting and fusing manner. In the selection step, this work newly designs two factors “Connectivity,” which measures how relevant a token is to another one, and “Dominance,” which measures the task relevance of tokens. With these two factors, it computes a representative score and then selects the representative token by using this score. In the fusion step, it utilizes connectivity to pair the less representative tokens with their representative counterparts, subsequently integrating the former into the latter. As a result, this work shows state-of-the-art performance.

**Strengths:**

- It achieves state-of-the-art efficiency without performance drops.
- Compared to prior token merging/pruning methods, this work effectively selects the representative token with global information and fuses neighborhood tokens.

**Weaknesses:**

The proposed method appears to be the 'select and fuse' type, which seems to require more computation than the existing methods, and I can't figure out at which part the speed improvement has been made. I can't find how many percent of the tokens were pruned finally by selecting and fusing the tokens compared to other models. It would be better to report the token reduction ratio and compare it with those of SoTA methods.

### Minor points
> presentation and writing

- In the Introduction, the explanation of the method proposed in the last paragraph seems to be cut off as if it were interrupted. For completeness, It needs to explain the matching and fusing part in the rest of the part.
- In page #7, “Distillation” paragraph is located among tables, which looks bad.

**Questions:**

The crucial reason for the faster speed compared to other models could it be because of the pruning of the total number of tokens? Or is it because the select-and-fuse calculation is simple and fast?